# DAnkrd49 and Bdbt act via Casein kinase Iε to regulate planar polarity in *Drosophila*

**Helen Strutt** ⬥*, **David Strutt** *

Department of Biomedical Science, University of Sheffield, Western Bank, Sheffield, United Kingdom

* h.strutt@sheffield.ac.uk (HS); d.strutt@sheffield.ac.uk (DS)

**Data Availability Statement:** All relevant data are within the manuscript and its Supporting Information files.

**Funding:** The work was funded by Wellcome Trust Senior Fellowship awards (https://wellcome.ac.uk, 084469/Z/07/Z, 100986/Z/13/Z and 210630/Z/18/

## Abstract

The core planar polarity proteins are essential mediators of tissue morphogenesis, controlling both the polarised production of cellular structures and polarised tissue movements. During development the core proteins promote planar polarisation by becoming asymmetrically localised to opposite cell edges within epithelial tissues, forming intercellular protein complexes that coordinate polarity between adjacent cells. Here we describe a novel protein complex that regulates the asymmetric localisation of the core proteins in the *Drosophila* pupal wing. DAnkrd49 (an ankyrin repeat protein) and Bride of Doubletime (Bdbt, a non-canonical FK506 binding protein family member) physically interact, and regulate each other's levels *in vivo*. Loss of either protein results in a reduction in core protein asymmetry and disruption of the placement of trichomes at the distal edge of pupal wing cells. Post-translational modifications are thought to be important for the regulation of core protein behaviour and their sorting to opposite cell edges. Consistent with this, we find that loss of DAnkrd49 or Bdbt leads to reduced phosphorylation of the core protein Dishevelled and to decreased Dishevelled levels both at cell junctions and in the cytoplasm. Bdbt has previously been shown to regulate activity of the kinase Discs Overgrown (Dco, also known as Doubletime or Casein Kinase Iε), and Dco itself has been implicated in regulating planar polarity by phosphorylating Dsh as well as the core protein Strabismus. We demonstrate that DAnkrd49 and Bdbt act as dominant suppressors of Dco activity. These findings support a model whereby Bdbt and DAnkrd49 act together to modulate the activity of Dco during planar polarity establishment.

## Author summary

In many animal tissues, sheets of cells are polarised in the plane of the tissue, which is evident by the production of polarised structures, such as hairs on the fly wing that point in the same direction or cilia that beat in the same direction. One group of proteins controlling this coordinated polarity are the core planar polarity proteins, which localise asymmetrically within cells such that some core proteins localise to one cell end and others to the opposite cell end. It is thought that modifications such as phosphorylation may locally regulate core protein stability, and this promotes sorting of proteins to different cell ends. We identify two proteins, DAnkrd49 and Bdbt, that form a complex and regulate core

Z) to D.S. The funders had no role in study design, data collection and analysis, decision to publish, or preparation of the manuscript.

**Competing interests:** The authors have declared that no competing interests exist.

protein asymmetry. Loss of either protein causes a reduction in overall levels of the core protein Dishevelled (Dsh), and a reduction in its phosphorylation. We provide evidence that the effect on core protein asymmetry is mediated via regulation of the kinase activity of Discs overgrown (Dco, also known as Doubletime/Casein Kinase Iε) by DAnkrd49 and Bdbt. We propose that modulation of Dco activity by DAnkrd49 and Bdbt is a key step in the sorting of core proteins to opposite cell ends.

## Introduction

Planar polarity describes the phenomenon whereby cells coordinate their polarity in the plane of a tissue: for example the hairs on the skin point in the same direction, cilia coordinate their beating, and cells coordinate their movements during tissue morphogenesis [1–3]. Understanding the mechanisms by which this coordinated polarisation occurs is of prime importance, as disruption of polarity can have diverse consequences, including neural tube closure defects, hydrocephalus and defects in neuronal migration [3, 4].

The fly wing is a well-characterised model system in which to study planar polarity. Each cell within the adult wing produces a single hair, or trichome, which points towards the distal end of the wing. Furthermore, viable mutations that cause characteristic swirling of the trichomes have been identified, and the genes associated with these mutations were subsequently found to be highly conserved, and to regulate planar polarity throughout the animal kingdom [5].

The core planar polarity proteins (hereafter known as the core proteins) are the best characterised group of proteins that regulate planar polarity. In the pupal wing, the core proteins adopt asymmetric subcellular localisations prior to trichome emergence, and in their absence trichomes emerge from the centre of the cell rather than at the distal cell edge [6]. The core proteins comprise the atypical cadherin Flamingo (Fmi, also known as Starry Night [Stan]), the transmembrane proteins Frizzled (Fz) and Strabismus (Stbm, also known as Van Gogh [Vang]), and three cytoplasmic proteins Dishevelled (Dsh), Prickle (Pk) and Diego (Dgo). Fmi localises to proximal and distal cell edges in the pupal wing, but is excluded from lateral cell edges, while Fz, Dsh and Dgo localise to distal cell edges and Stbm and Pk to proximal cell edges (Fig 1A). These proteins form intercellular complexes at cell junctions, that couple neighbouring cells and allow them to coordinate their polarity [1, 2].

The mechanisms by which the core proteins become asymmetrically localised are poorly understood. The overall direction of polarity is thought to be determined by tissue-specific 'global' cues: these may include gradients of morphogens or Fat/Dachsous cadherin activity, or other cues such as mechanical tension [7]. In the wing these global cues may directly regulate core protein localisation or act indirectly via effects on growth and tissue morphogenesis [7]. Global cues are thought to lead to subtle biases in core protein localisation within cells that are subsequently amplified by feedback between the core proteins, in which positive (stabilising) interactions between complexes of the same orientation are coupled with negative (destabilising) interactions between complexes of opposite orientation (Fig 1A). In mathematical models, such feedback interactions have been demonstrated to be sufficient to amplify weak biases in protein localisation, leading to sorting of complexes and robust asymmetry (e.g. [8–11]).

Experimental evidence for feedback is only beginning to be elucidated. Cell culture experiments have demonstrated competitive binding between several of the core proteins, which may be important for feedback [8, 12–14]. Furthermore, we have recently shown that the core protein Pk acts through Dsh to destabilise Fz in the same cell, while stabilising Fz across cell junctions via Stbm [15].

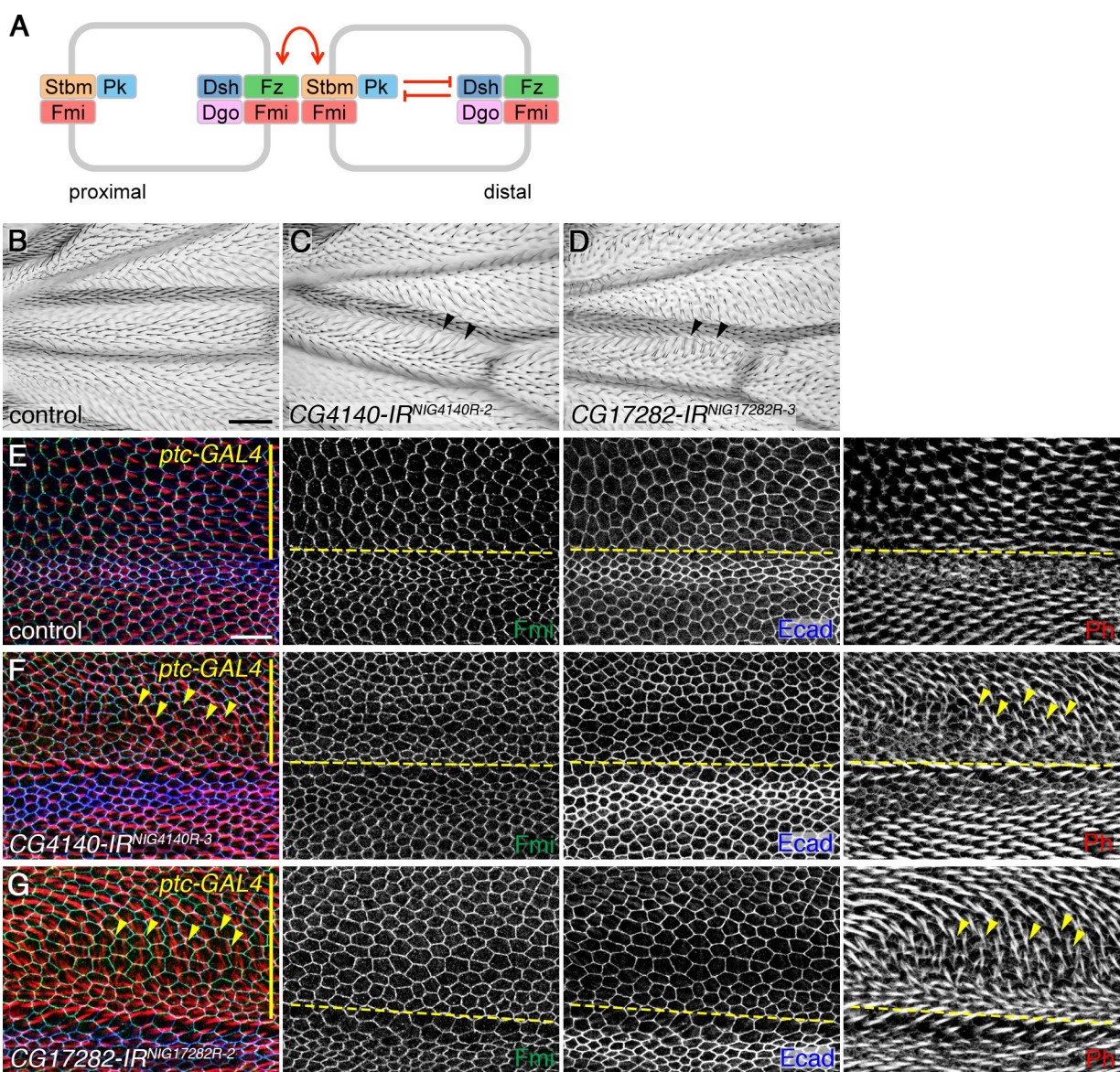

**Fig 1. RNAi knockdown of *DAnkrd49* and *Bdbt* disrupts planar polarity.** (A) Diagram to illustrate asymmetric localisation of the core planar polarity proteins. Fz, Dsh and Dgo localise to distal cell ends, while Stbm and Pk localise proximally and Fmi localises both proximally and distally. Feedback interactions are thought to lead to mutual inhibition between proximal and distal complex components (red inhibitory lines), while intercellular interactions between proximal and distal components are thought to stabilise asymmetric complexes (red arrows). (B-D) Dorsal surface of adult male wings from control wild-type flies (B), or from flies expressing RNAi against *CG4140/DAnkrd49* (C, NIG line *4140R-2*), or *CG17282/Bdbt* (D, NIG line *17282R-3*), under control of the *MS1096-GAL4* driver at 25˚C. Distal is to the right and anterior is up in these and all other images. Black arrowheads mark a proximal region of the wing, between veins 3 and 4, in which trichomes point distally in wild-type wings, but swirl anteriorly in *DAnkrd49* and *Bdbt* knockdown wings. Note that anterior to vein 3, trichomes tend to point towards the vein in wild-type as well as in mutants wings. Scale bar 50 μm. (E-G) Pupal wings expressing a control RNAi (E, VDRC line *39864*, targeting *Sik1*, a gene unrelated to planar polarity), or RNAi against *CG4140/DAnkrd49* (F, NIG line *4140R-3*), or *CG17282/Bdbt* (G, NIG line *17282R-2*), under control of the *ptc-GAL4* driver. Larvae were raised at 25˚C, collected as white prepupae and then aged for 27hr at 29˚C. Wings immunolabelled for Fmi (green) and Ecad (blue), and trichomes labelled with Phalloidin (red). *ptc-GAL4* is expressed at the top of the image (yellow bar), and the yellow dotted line marks the anterior-posterior boundary. Yellow arrowheads mark some trichomes which emerge from posterior cell edges, rather than from distal cell edges as normally. Scale bar 10 μm.

In order for feedback to operate, cells must utilise the general cellular machinery: for example active endocytosis is necessary for Pk to destabilise Fz [15]. Furthermore, post-translational modifications of the core proteins are likely to be key mediators of feedback. For example loss

of ubiquitination pathway components and some protein kinases have been shown to disrupt planar polarity. In flies, a Cullin-3/Diablo/Kelch ubiquitin ligase complex regulates Dsh levels at cell junctions, while the de-ubiquitinase Fat Facets regulates Fmi levels [16]. Stbm also negatively regulates Pk levels, and ubiquitination of Pk by Cullin-1/SkpA/Slimb promotes internalisation of Fmi-Stbm-Pk complexes [17–19]. Similarly, in vertebrates, the Stbm homologue Vangl2 may promote local degradation of Pk via ubiquitination by Smurf E3 ubiquitin ligases [20, 21]. Furthermore, *Drosophila* Fz phosphorylation is mediated by atypical Protein Kinase C [22], and Dsh is a target of phosphorylation by the Discs Overgrown (Dco, also known as Doubletime [Dbt] or Casein Kinase Iε [CKIε]) and Abelson kinases [23–25]. Dco/CKIε has also been implicated in phosphorylation of Stbm in both flies and vertebrates [26–29].

Here we describe the identification of two new regulators of planar polarity in the *Drosophila* wing. We show that Bride of Doubletime (Bdbt) and DAnkrd49 are binding partners that regulate each other's levels. Loss of either protein disrupts asymmetric localisation of the core proteins. Furthermore, they regulate overall levels of Dsh and Dsh phosphorylation in the pupal wing, and we provide evidence that they act by modulating the activity of the kinase Dco.

## Results

### Identification of two novel regulators of planar polarity

To identify novel regulators of planar polarity in *Drosophila*, we performed an unbiased RNAi screen, using the *MS1096-GAL4* driver to express RNAi lines throughout the developing wing blade and examining the adult wings for evidence of planar polarity defects in trichome (wing hair) orientation (see also [18, 30]). We identified two RNAi lines which gave a similar phenotype, in which trichomes were misoriented in the proximal region of the wing (Fig 1B–1D). These RNAi lines targeted two different loci: *CG4140* and *CG17282*. Additional RNAi lines corresponding to these loci were obtained, and when screened using *MS1096-GAL4*, or *ptc-GAL4* in the presence of *UAS-Dcr2* (S1 Table), gave qualitatively similar trichome planar polarity defects.

To confirm that these phenotypes were due to mislocalisation of the core polarity proteins, we examined pupal wings in which RNAi constructs were expressed along the anterior-posterior compartment boundary using *ptc-GAL4*. As in adult wings (S1A–S1C Fig), trichome polarity was disrupted, with trichomes emerging from incorrect cell edges and swirling towards the anterior-posterior boundary (Fig 1E–1G). Trichome initiation was also delayed within the *ptc-GAL4* domain, and asymmetric localisation of the core protein Fmi was disrupted, such that it showed a more uniform distribution around the apical junctions (Fig 1E–1G and S1D–S1F Fig). This loss of asymmetry was not caused by a general defect in membrane organisation or apical-basal polarity, as E-cadherin localisation was not affected. However an increase in cell size was evident.

*CG4140* (also known as *l(2)35Be*) encodes a 215 amino acid protein with ankyrin repeats (Fig 2A). It is a homologue of the poorly characterised human gene Ankyrin Repeat Domain-Containing Protein 49 (ANKRD49, S2A Fig), and we will hereafter refer to the fly gene as *DAnkrd49*. *CG17282* is also known as *Bride of Doubletime* (*Bdbt*), and encodes a member of the FKBP (FK506 binding protein) family. Bdbt has 3 tetratricopeptide repeats (Fig 2A) that may mediate protein-protein interactions. Like other FKBP family members, its N-terminus has structural similarity to peptidyl prolyl isomerases (PPIases), but it lacks critical residues for catalytic activity [31]. Interestingly, its N-terminus binds to the kinase Dco: loss of Bdbt in adult flies promotes hyperphosphorylation of Dco, and reduces its ability to phosphorylate target proteins (Fig 2B) [31, 32].

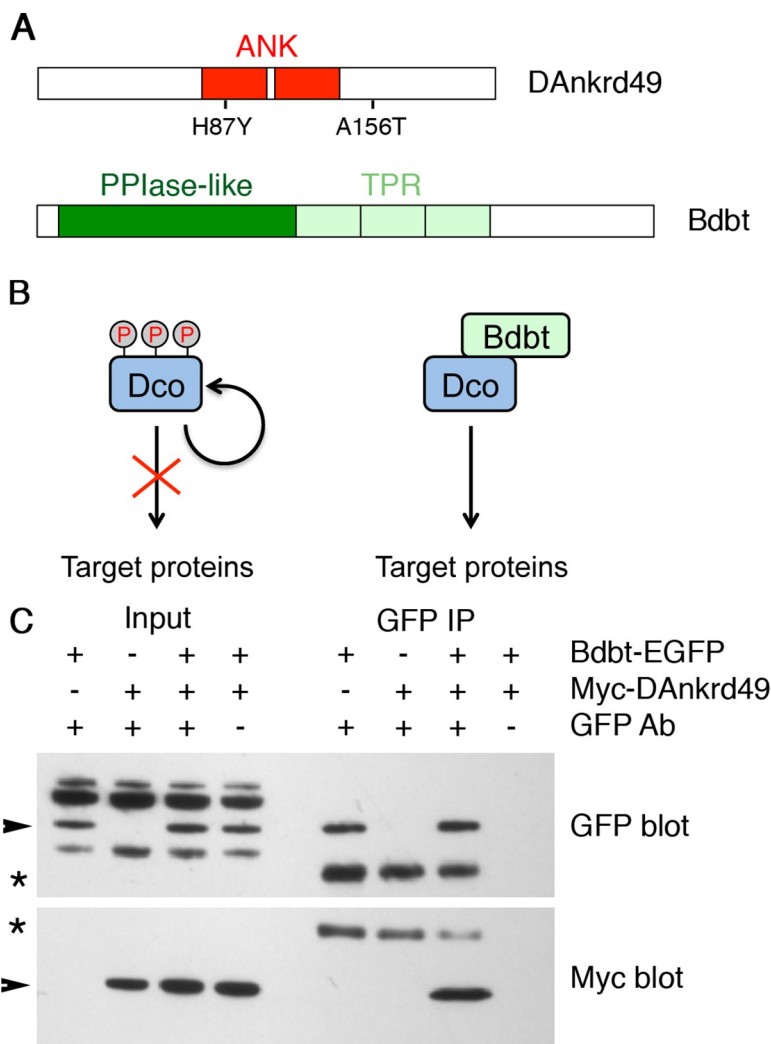

**Fig 2. Physical interaction between DAnkrd49 and Bdbt** (A) Schematics of DAnkrd49 (top) and Bdbt (bottom) protein structures. The ankyrin repeats of DAnkrd49 are shown in red, and the positions of the two point mutations seen in the *DAnkrd49$^{l(2)35Be4}$* allele are indicated. The peptidyl prolyl isomerase-like domain of Bdbt is shown in dark green and the tetratricopeptide repeats in pale green. (B) Regulation of Dco kinase by Bdbt, based on Fan et al 2013 [31]. In the absence of Bdbt, Dco is hyperphosphorylated and has reduced kinase activity on its target proteins [42]. Binding of Bdbt to Dco prevents hyperphosphorylation of Dco, and leads to increased Dco kinase activity. (C) Western blot showing co-immunoprecipitation of Myc-DAnkrd49 by Bdbt-EGFP, after co-transfection into S2 cells. Arrowheads indicate specific bands and asterisks indicate antibody heavy chains.

## Physical interaction between Bdbt and DAnkrd49

Bdbt and DAnkrd49 were identified as binding partners in an early release of the *Drosophila* Protein Interaction Mapping project [33], but were not considered to be high confidence interactors in the published analysis (DPIM-2) [34]. In order to confirm this interaction, EGFP-tagged Bdbt and Myc-tagged DAnkrd49 were co-transfected into *Drosophila* S2 cells. Immunoprecipitation with GFP antibody led to pulldown of Myc-DAnkrd49 (Fig 2C). This suggests that Bdbt and DAnkrD49 interact and act as a complex to regulate core protein asymmetry.

### *DAnkrd49* and *Bdbt* mutants cause loss of core protein asymmetry

To confirm that the RNAi phenotypes were due to knockdown of the expected target genes, we examined loss of function mutations of *DAnkrd49* and *Bdbt*. EMS-induced alleles of *DAnkrd49* have been identified, which are homozygous lethal, but otherwise uncharacterised [35]. Sequencing of one of these alleles, *DAnkrd49*[l(2)35Be4], revealed H87Y and A156T mutations (Fig 2A and S2A Fig). H87 is highly conserved in all species, and mutation to Y is predicted to be highly deleterious, while A156 is less well conserved, and mutation to T predicted to be tolerated (SIFT analysis) [36]. Mutant clones of *DAnkrd49*[l(2)35Be4] were generated in the pupal wing, which revealed reduced asymmetric localisation of Fmi (Fig 3A and 3C),

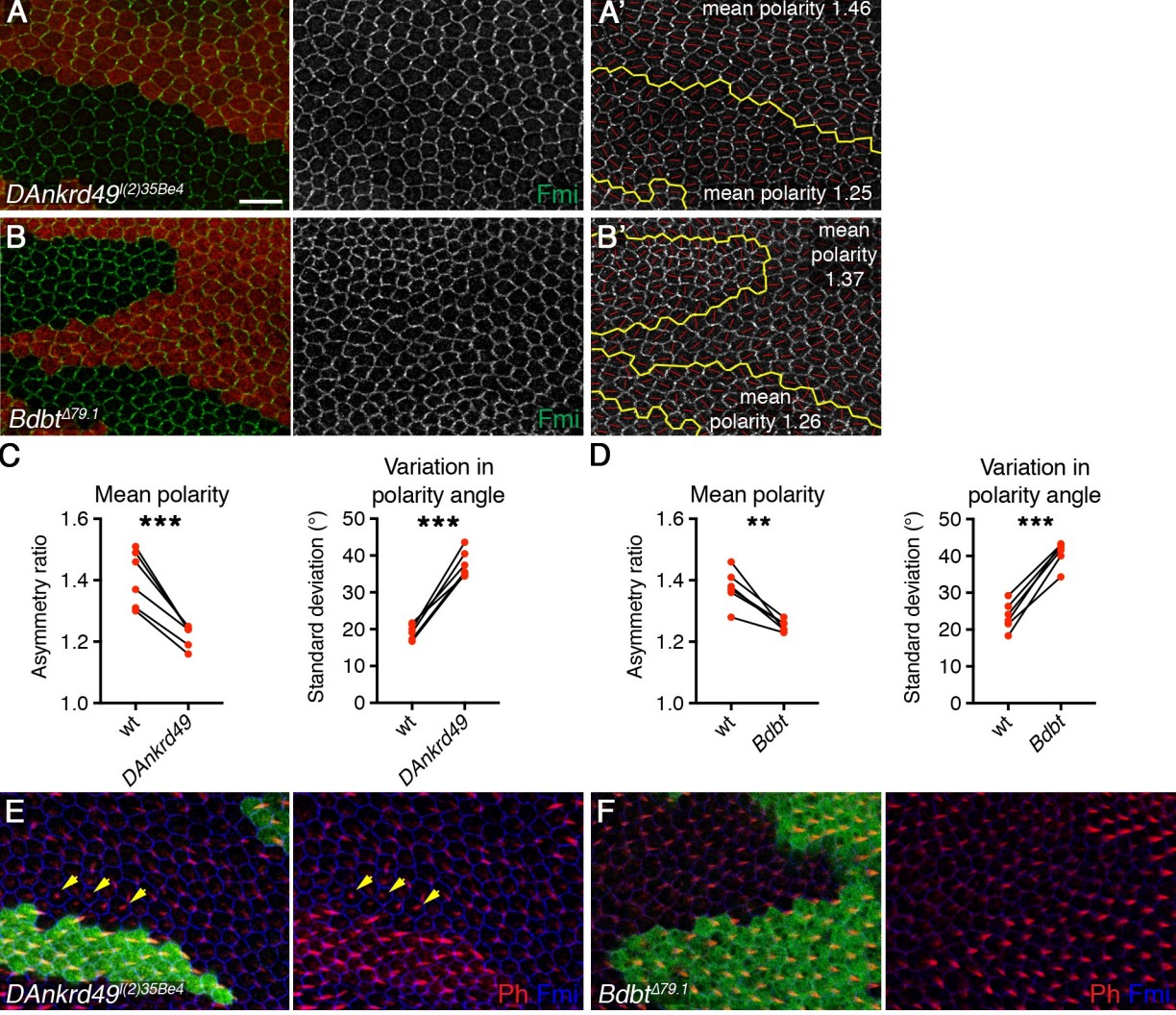

**Fig 3. DAnkrd49 and Bdbt regulate asymmetric localisation of core proteins.** (A,B) 28hr APF pupal wings carrying clones of cells lacking *DAnkrd49* (*DAnkrd49*[l(2)35Be4], A), or *Bdbt* (*Bdbt*[Δ79.1], B), marked by loss of GFP or β-gal respectively (red). Wings immunolabelled for Fmi (green). Scale bar 10 μm. (A',B') Clones in A and B showing the clone outline (yellow line) and the polarity nematic for each cell (red lines). Length of red line (polarity magnitude) is reduced in *DAnkrd49* and *Bdbt* clones compared to wild-type tissue. (C,D) Quantitation of mean polarity and variation in polarity angle, in wild-type and *DAnkrd49* (C) or *Bdbt* (D) mutant tissue, values from the same wing are linked by black bars. Wild-type tissue was quantitated several cells away from clone boundaries, due to some disruption in polarity on clone boundaries. Paired t-tests were used to compare values in the same wing, **p<0.01, ***p<0.001. 6 wings of each genotype were quantified. See S2 Table for numerical data. (E,F) 32.5hr APF pupal wings carrying clones of cells lacking *DAnkrd49* (*DAnkrd49*[l(2)35Be4], E), or *Bdbt* (*Bdbt*[Δ79.1], F), marked by loss of GFP or β-gal respectively (green). Wings immunolabelled for Fmi (blue) and labelled with Phalloidin (red). Yellow arrows indicate trichomes which are delayed and initiate in the cell centre rather than at the cell edge.

accompanied by delayed trichome initiation, and formation of trichomes in the cell centre rather than at the distal cell edge (Fig 3E). Levels of the junctional protein Armadillo (Arm) were not affected in these clones (S3A Fig), suggesting there is no general defect in junctional integrity. When small patches of wild-type cells were surrounded by $DAnkrd49^{l(2)35Be4}$ mutant cells, polarity of the wild-type cells was often slightly disrupted. However, we did not see consistent non-autonomous effects on proximal or distal clone boundaries.

As $DAnkrd49^{l(2)35Be4}$ is unlikely to be a null mutation, and the chromosome also carries additional recessive markers, we made an independent allele by deleting the entire gene by homologous recombination (S2B Fig, see also Materials and Methods). Multiple independent alleles were generated, all of which were homozygous lethal, and lethal when transheterozygous with $DAnkrd49^{l(2)35Be4}$. We then made mutant clones of $DAnkrd49^{AD3.1}$; however few clones were recovered, and most clones that were seen consisted of only a few cells, suggesting a proliferation defect. However rare clones that were slightly larger gave a qualitatively similar polarity phenotype to $DAnkrd49^{l(2)35Be4}$ mutant clones, with reduced core protein asymmetry and delayed trichome formation (S3D and S3F Fig). Due to the poor proliferation of $DAnkrd49^{AD3.1}$ mutant clones, we used the $DAnkrd49^{l(2)35Be4}$ hypomorph for subsequent experiments.

No mutant alleles were available for *Bdbt*, so we again used homologous recombination to knock out the entire coding region (S2B Fig, see also Materials and Methods). A single mutant allele was generated, which was homozygous lethal and failed to complement deficiencies for the region. In pupal wing clones, *Bdbt* mutations gave a similar phenotype to *DAnkrd49* mutations, notably reduced Fmi asymmetry and delayed trichome initiation (Fig 3B, 3D and 3F). In some clones we saw a disruption in tissue structure, as seen by reduced Armadillo (Arm) staining at junctions (S3C Fig). However core protein asymmetry was reduced even in clones with normal Arm staining (S3B Fig). No obvious effect on cell proliferation was seen, unlike in *DAnkrd49* clones. This suggests that DAnkrd49 may have additional functions, independent of Bdbt, or that differential perdurance of the two gene products within mutant clones may cause different effects on cell growth and viability.

## EGFP-tagged DAnkrd49 and Bdbt localise uniformly in the cytoplasm and DAnkrd49 and Bdbt regulate each other's levels

Planar polarity proteins belonging to the 'core' localise asymmetrically within the pupal wing, and loss of any of these proteins disrupts the localisation of the others. In contrast, other regulatory proteins such as kinases regulate the asymmetry of the core proteins but have not been observed to themselves localise asymmetrically (e.g. [16, 24]). To determine to which class DAnkrd49 and Bdbt belong, we expressed EGFP-tagged DAnkrd49 and Bdbt in the pupal wing, under control of the ubiquitous *Act5C* promoter. In both cases we saw uniform expression in the cytoplasm, with some localisation to the plasma membrane; but no asymmetric localisation (Fig 4A and 4B). Notably, the transgenes rescued their respective mutant phenotypes (Fig 4A–4D, compare with Fig 3A–3D), confirming both that the transgenes are functional and that the mutant phenotype is associated with mutations in the expected gene locus. Cellular levels of DAnkrd49-EGFP were increased in the absence of endogenous DAnkrd49.

As our co-immunoprecipitation experiments suggest that DAnkrd49 and Bdbt act as a complex, we examined whether they regulated each other's localisation or levels. Interestingly, loss of *Bdbt* causes a reduction in overall levels of DAnkrd49-EGFP in the pupal wing, and vice versa (Fig 4E and 4F). This is consistent with a model in which DAnkrd49 and Bdbt act in a complex in which they promote each other's stability.

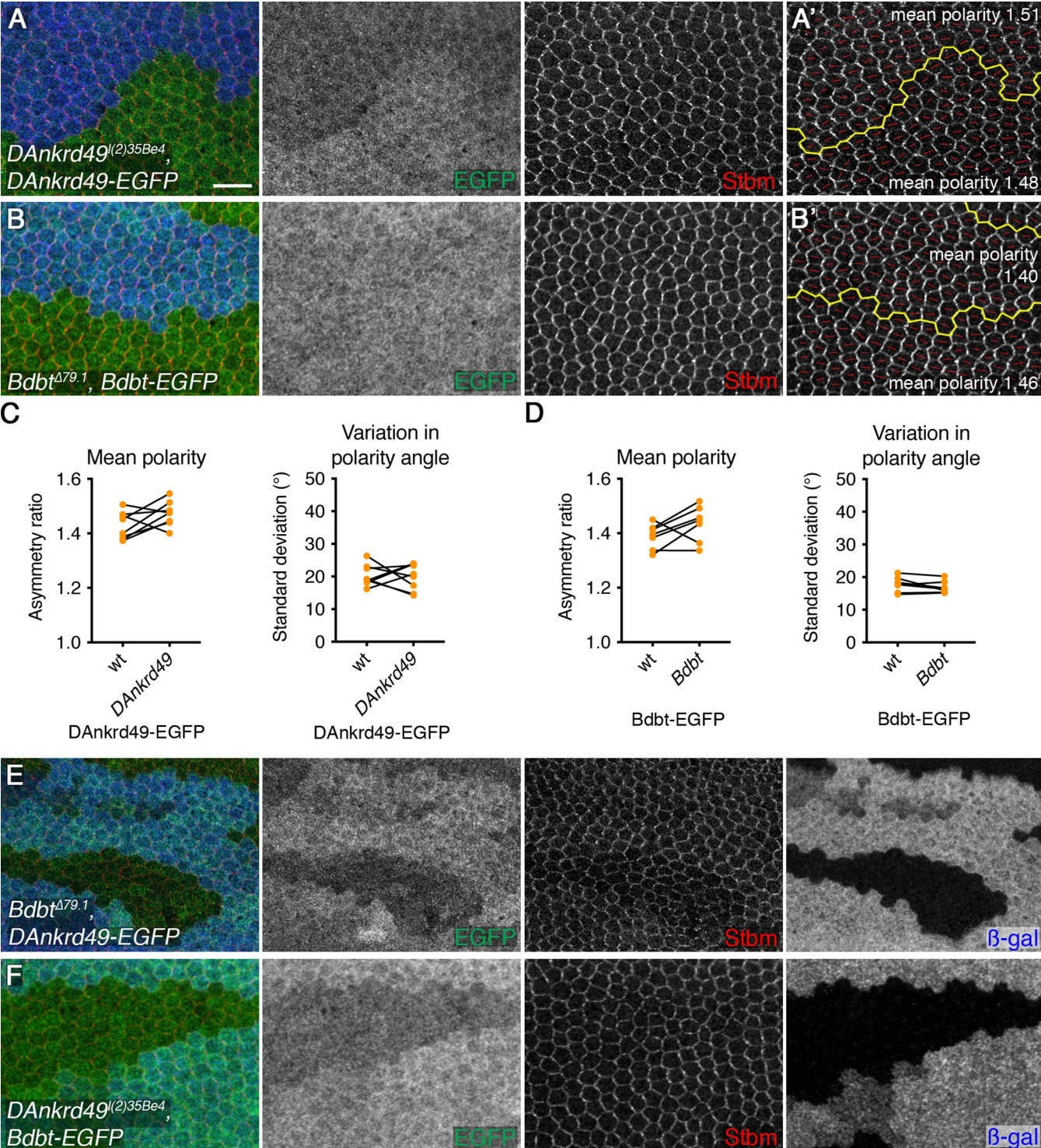

**Fig 4. DAnkrd49 and Bdbt mutually regulate each other's protein levels.** (A,B) 28hr APF pupal wings carrying clones of cells lacking *DAnkrd49* (*DAnkrd49^l (2)35Be4*) in a background expressing *ActP-DAnkrd49-EGFP* (A), or clones of cells lacking *Bdbt* (*Bdbt^Δ79.1*) in a background expressing *ActP-Bdbt-EGFP* (B). Wings immunolabelled for GFP (green) and Stbm (red). Clones marked by loss of β-gal (blue). Note that levels of DAnkrd49-EGFP are slightly increased in the absence of endogenous *DAnkrd49*. Scale bar 10 μm. (A',B') Clones in A and B showing the clone outline (yellow line) and the polarity nematic for each cell (red lines). Length of red line (polarity magnitude) is not significantly different in wild-type tissue and inside the *DAnkrd49* and *Bdbt* clones, showing that the mutant phenotype is rescued by DAnkrd49-EGFP and Bdbt-EGFP respectively. (C,D) Quantitation of mean polarity and variation in polarity angle, in clones of *DAnkrd49* in a background expressing *ActP-DAnkrd49-EGFP* (C) or clones of *Bdbt* in a background expressing *ActP-Bdbt-EGFP* (D), values from clone and

non-clone regions of the same wing are linked by black bars. Paired t-tests were used to compare values in the same wing, no significant differences were found. 8 wings were quantified in (C) and 7 wings were quantified in (D). See S2 Table for numerical data. (E,F) 28hr APF pupal wings carrying clones of cells lacking *Bdbt* (*Bdbt^{A79.1}*) in a background expressing *ActP-DAnkrd49-EGFP* (E), or clones of cells lacking *DAnkrd49* (*DAnkrd49^{l(2)35Be4}*) in a background expressing *ActP-Bdbt-EGFP* (F). Wings immunolabelled for GFP (green) and Stbm (red). Clones marked by loss of β-gal (blue). Note that levels of the EGFP-tagged proteins are reduced within the clones.

## DAnkrd49 and Bdbt regulate overall levels of Dsh

To determine the basis for the loss of core protein asymmetry, we further investigated the effects of loss of *DAnkrd49* and *Bdbt* on core protein localisation. For mutations in both genes, we found that levels of Dsh at junctions were severely reduced (Fig 5A and 5B and S3E Fig), while we did not observe changes in levels of any other core proteins we examined (S3A, S3B Fig and S4A–S4D Fig). In particular, levels of Fz, which recruits Dsh to junctions, were similar to wild-type (S4A and S4B Fig). As our Dsh antibody gives variable amounts of background staining, we confirmed this result using a *P[acman]-EGFP-dsh* rescue construct. This revealed a reduction in Dsh levels, not only at the junctions but also in the cytoplasm (Fig 5C and 5D). Levels of Dsh-ECFP expressed under a heterologous *arm* promoter were also reduced, consistent with the effects being post-transcriptional (S4E and S4F Fig). Furthermore Dsh levels were rescued in *DAnkrd49 and Bdbt* mutant clones when DAnkrd49-EGFP and Bdbt-EGFP were expressed, respectively (S4G and S4H Fig).

To confirm that overall cellular levels of Dsh are reduced in *DAnkrd49* and *Bdbt* mutants, we carried out western blots on pupal wing extracts. As loss-of-function mutants were lethal, we expressed RNAi in the pupal wing using the *MS1096-GAL4* driver. Expressing RNAi against *Bdbt* caused variable lethality (see S1 Table) and insufficient pupal wings were obtained for western analysis. However, expression of RNAi against *DAnkrd49* did not affect pupal wing development, and a clear decrease in cellular levels of Dsh on western blots was observed (Fig 5E and 5F).

## Bdbt and DAnkrd49 regulate levels of Dsh phosphorylation

Bdbt is a non-canonical member of the FKBP family, which lacks the critical residues for PPIase activity [31]. However, FKBP family members can also function as molecular chaperones, assisting in protein folding independent of PPIase activity [37, 38]. One possibility therefore is that Bdbt and DAnkrd49 act as molecular chaperones, directly promoting Dsh stability. To investigate this, we expressed Myc-tagged Bdbt or DAnkrd49 in S2 cells, and attempted to co-immunoprecipitate Dsh-ECFP. No co-immunoprecipitation was seen (S5 Fig): furthermore no interaction was seen in the genome-wide interaction screen that found the Bdbt-DAnkrd49 interaction [33, 34]. Thus we cannot find evidence to support a chaperone model.

An alternative model is suggested by the fact that Bdbt physically interacts with Dco [31, 32], and that Dco is known to regulate Dsh phosphorylation in both canonical Wnt signalling and in planar polarity [23, 24, 39–41]. Dco can autophosphorylate its own C-terminus, and this autophosphorylation reduces its activity [42]. Deletion of the C-terminus of Dco increases Dco activity in *in vitro* assays, consistent with it being an inhibitory domain [42]. Binding of Bdbt to Dco is dependent on the C-terminus of Dco, and loss of Bdbt causes hyperphosphorylation of Dco and reduced Dco activity *in vivo* (Fig 2B) [31]. This leads to an attractive model in which Bdbt and DAnkrd49 regulate Dsh indirectly, through regulation of Dco activity. Consistent with this, we see a reduction of relative levels of Dsh phosphorylation in pupal wings expressing RNAi against *DAnkrd49* (Fig 5G), phenocopying the reduced Dsh phosphorylation that was previously shown in *dco* hypomorphs [24].

Previously we reported a loss of asymmetry in *dco* hypomorphic mutant clones [24], but we did not report any effect on overall Dsh levels. However, upon re-examination of *dco* mutant clones, we saw a reduction in levels of Dsh at junctions in mutant tissue, while no reduction in

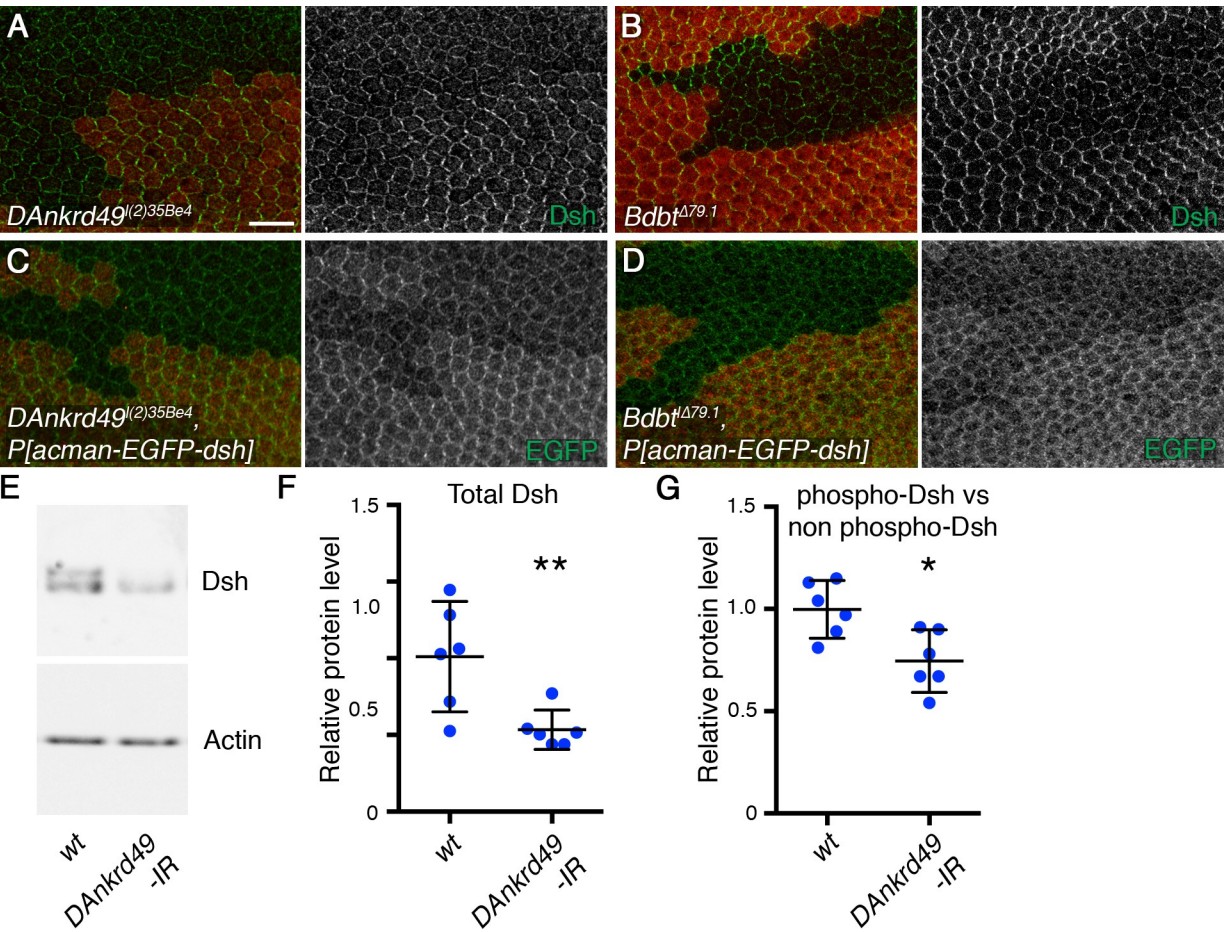

**Fig 5. DAnkrd49 and Bdbt regulate Dsh levels.** (A,B) 28hr APF pupal wings carrying clones of cells lacking *DAnkrd49* (*DAnkrd49^{l(2)35Be4}*), marked by loss of GFP (red, A), or lacking *Bdbt* (*Bdbt^{Δ79.1}*), marked by loss of β-gal (red, B). Wings immunolabelled for Dsh (green). Scale bar 10 μm. (C-D) 28hr APF pupal wings carrying clones of cells lacking *DAnkrd49* (C, *DAnkrd49^{l(2)35Be4}*), or clones of cells lacking *Bdbt* (D, *Bdbt^{Δ79.1}*), in a background expressing one copy of *P[acman]-EGFP-dsh*. Wings immunolabelled for GFP (green), clones marked by loss of β-gal (red). (E) Representative western blot from wild-type pupal wings, or pupal wings expressing RNAi against *DAnkrd49* (line *4140R-3*), under control of the *MS1096-GAL4* driver. Blot probed with Dsh (top) and Actin (bottom). Overall Dsh levels decrease in wings expressing *DAnkrd49* RNAi. (F) Quantitation of Dsh levels from western blotting, normalised to Actin, from 6 biological replicates. Error bars are 95% confidence intervals, $^{**}$p = 0.01 (unpaired t-test). See S2 Table for numerical data. (G) Comparison of levels of slow migrating Dsh (hyperphosphorylated) relative to fast migrating Dsh (less phosphorylated). The graph shows a ratio of the maximum intensity of the two bands averaged across the width of each band. Error bars are 95% confidence intervals, $^{*}$p = 0.011 (unpaired t-test). See S2 Table for numerical data.

other core proteins such as Fmi was seen (Fig 6A). Furthermore, when *P[acman]-EGFP-dsh* was expressed in pupal wings, a reduction in both junctional and cytoplasmic levels of Dsh was seen in *dco* mutant tissue (Fig 6B), again phenocopying *DAnkrd49* and *Bdbt*. Expression of dominant negative Dco has also been shown to reduce levels of *P[acman]-EGFP-dsh* [29].

## Bdbt and DAnkrd49 interact genetically with Dco

Overexpression of *dco* causes defects in trichome orientation in the adult wing [23, 24, 29, 41]. As DAnkrd49 and Bdbt are expected to promote Dco activity, to seek support for our model we investigated whether a reduction in their activity would suppress the *dco* overexpression phenotype. *dco* was overexpressed in the posterior compartment of the wing using the *en-GAL4* driver, which reproducibly caused a large trichome swirl close to the posterior cross vein, and a smaller swirl in the distal region of the wing (Fig 7A and 7D). Interestingly, the size

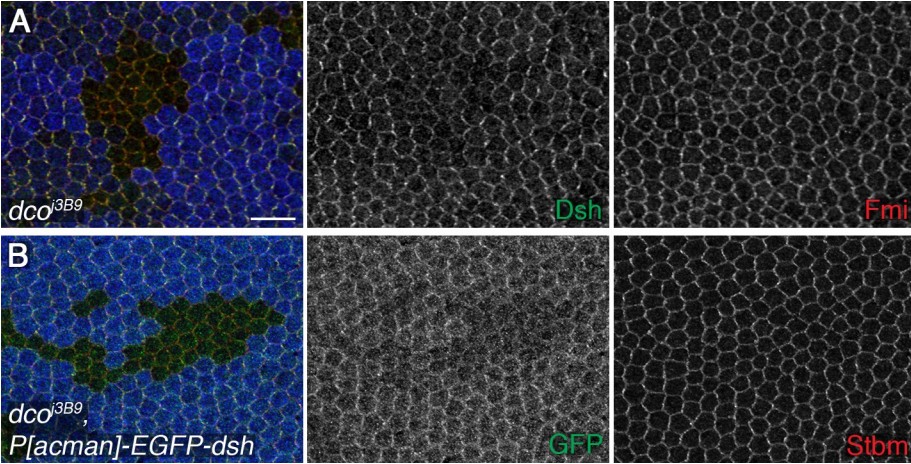

**Fig 6. Loss of Dco activity reduces Dsh levels.** (A) 28hr APF pupal wings carrying *dco* hypomorphic mutant clones of cells (*dco*$^{j3B9}$), marked by loss of β-gal (blue). Wings immunolabelled for Dsh (green) or Fmi (red). Scale bar 10 μm. (B) 28hr APF pupal wings carrying clones of cells lacking *dco* (*dco*$^{j3B9}$), in a background expressing one copy of *P[acman]-EGFP-dsh*. Wings immunolabelled for GFP (green), or Stbm (red), and clones marked by loss of β-gal (blue).

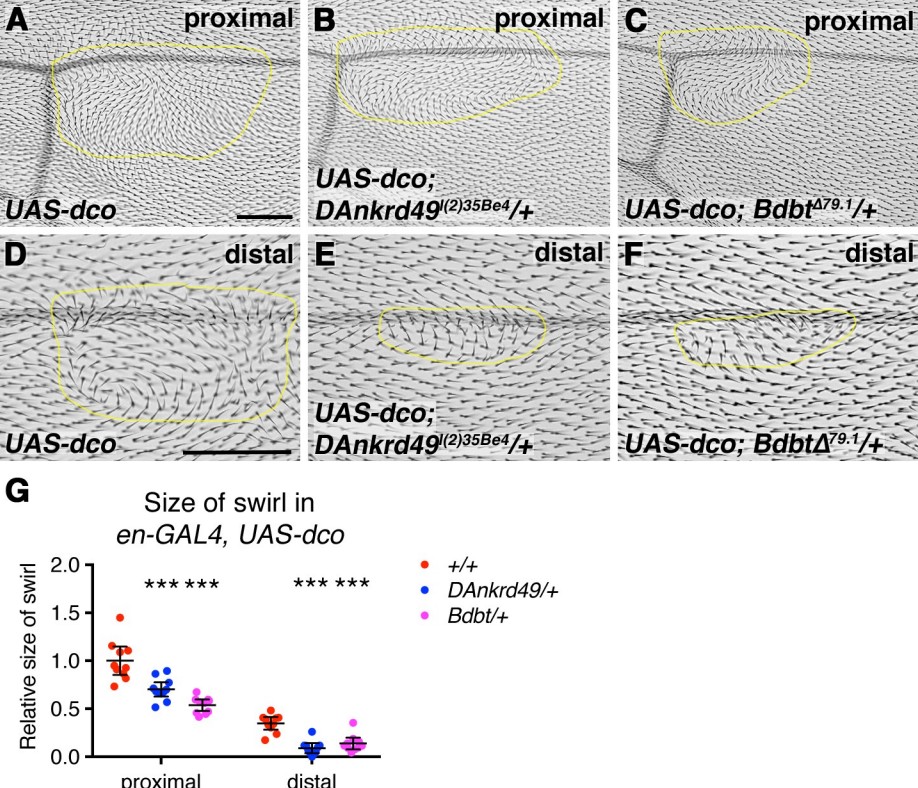

**Fig 7. Genetic interactions between *dco*, *DAnkrd49* and *Bdbt*.** (A-F) Dorsal surface of adult wings from *UAS-dco/+; en-GAL4/+* (A,D), *UAS-dco/+; en-GAL4/DAnkrd49*$^{l(2)35Be4}$ (B,E) and *UAS-dco/+; en-GAL4/+; Bdbt*$^{Δ79.1/+}$ (C,F) female flies, raised at 25˚C. (A-C) Images near the posterior cross vein. (D-F) Images around vein 4 near the distal tip of the wing. The yellow line marks the area of wing in which trichomes swirl. Scale bars 100 μm. (G) Quantitation of the area of wing in which trichomes swirl, in 10 wings of the genotypes from A-F. Samples were compared to *UAS-dco/+; en-GAL4/+* using one-way ANOVA with Dunnett's multiple comparisons test, \*\*\*p≤0.001. Error bars are 95% confidence intervals. See S2 Table for numerical data.

of the proximal swirl was reduced, and the distal swirl was almost entirely abolished in flies heterozygous for either *DAnkrd49* or *Bdbt* (Fig 7B, 7C and 7E–7G).

Overall the genetic interactions between *dco* and *DAnkrd49* and *Bdbt* support them acting in a common pathway. Coupled with the phenotypic similarities, we propose that DAnkrd49 and Bdbt regulate Dsh levels via modulating Dco activity.

## Discussion

Asymmetric localisation of the core proteins depends on the tight regulation of protein dynamics and stability. Current models suggest that in order for core protein complexes to become aligned in the same direction, complexes must destabilise each other when they are in opposite orientations. In contrast when complexes are in the same orientation, they become more stable. Understanding how core protein stability is regulated is therefore of key importance.

Here we present evidence for a novel protein complex regulating planar polarity, consisting of the ankyrin repeat protein DAnkrd49 and the non-canonical FKBP family member Bdbt. DAnkrd49 and Bdbt physically interact *in vitro* and regulate each other's levels *in vivo*. This suggests that they may act in a complex in which each is required to stabilise the other, a model supported by the phenotypic similarity in mutant clones. We show that their activity is required for core protein asymmetry in the pupal wing, and for normal levels and phosphorylation of the cytoplasmic core protein Dsh.

Previous work on circadian rhythms has established a physical interaction between Bdbt and the kinase Dco, and that Dco activity is regulated by Bdbt [31]. Notably, Dco has previously been implicated in promoting both Dsh and Stbm phosphorylation in planar polarity signalling [23, 24, 26–29]. Thus, our data are consistent with both DAnkrd49 and Bdbt acting to regulate Dco activity in planar polarity. Firstly, loss of function clones of *dco* and *DAnkrd49/Bdbt* share a common phenotype: in their absence we see reduced overall levels of Dsh and reduced levels of Dsh phosphorylation (this work and [24]). Secondly, we see strong genetic interactions between *dco* and *DAnkrd49/Bdbt*. Although a physical interaction has been observed between Dco and Bdbt [31], we do not see a direct interaction between Dco and DAnkrd49. Therefore, a simple model is that the role of DAnkrd49 is to stabilise Bdbt, while Bdbt directly regulates Dco activity. Thus we define a regulatory cascade, whereby DAnkrd49 and Bdbt promote Dco activity, which in turn regulates phosphorylation of core proteins and asymmetric localisation.

An alternative model is that DAnkrd49 and Bdbt act to stabilise Dsh, independently of Dco. Proteins of the FKBP family are known to regulate the stability of target proteins [37, 38]. In canonical FKBP family members this stabilisation is a result of PPIase activity, which assists protein folding; whereas other family members stabilise their target proteins by direct binding [37]. The catalytic sites for PPIase are not conserved in Bdbt [31]. We were also unable to see a binding interaction between Dsh and either DAnkrd49 or Bdbt, arguing that Bdbt is unlikely to directly stabilise Dsh.

In addition to defects in planar polarity, we also see other pleiotropic defects in *DAnkrd49* clones and *Bdbt* clones, including cell size defects, reduced proliferation and poor viability. These could plausibly be explained by regulation of Dco activity by DAnkrd49 and Bdbt. Dco acts in multiple signalling pathways. It phosphorylates the tumour suppressor Fat, which regulates cell growth and survival via the Hippo signalling pathway [43–47]. Interestingly, a hypomorphic mutation in *dco* causes tissue overgrowth and increased activity of the caspase inhibitor DIAP1, phenocopying loss of Hippo signalling pathway components [43, 44], while cells completely lacking Dco activity have reduced expression of the caspase inhibitor DIAP1 and reduced proliferation [45]. Dco may also act in additional signalling pathways, and these

are thought to include Hedgehog signalling and canonical Wnt signalling [48]. Hence, the multitude of signalling pathways regulated by Dco may explain the complex phenotypes seen in the absence of DAnkrd49 and Bdbt. Alternatively, it is possible that DAnkrd49 and Bdbt may regulate other downstream targets in addition to Dco.

As Dco has also been implicated in phosphorylating Stbm [26–28], loss of DAnkrd49 and Bdbt is also expected to reduce Stbm phosphorylation in pupal wings. However we have not been able to verify this directly, as Stbm mobility in SDS-PAGE is only marginally increased when Dco activity is reduced [27, 29], and is not noticeably altered in extracts from animals with reduced activity of DAnkrd49 or Bdbt. Nevertheless, our recent work has shown that phosphorylation of both Stbm and Dsh by Dco is functionally important in establishing correct planar polarity [29], making the regulation of Dco activity of great interest.

Interestingly, protein interactions studies in human cell lines have identified a STRING network between Ankrd49, CKIδ/CKIε and FKBP family members [49–51]. Although the FKBP proteins identified in these studies are not the closest orthologues to *Drosophila* Bdbt, this nevertheless suggests conservation of a regulatory cascade of Ankrd49/FKBP promoting Dco activity, which in turn might phosphorylate core planar polarity proteins. Furthermore, the DISEASES online resource, which integrates results from text mining, manually curated disease-gene associations and genome-wide association studies, has linked Ankrd49 with brachydactyly subtypes (diseases.jensenlab.org) [52]. Brachydactyly is a key feature of Robinow syndrome, a disease closely associated with core planar polarity mutations [53]; thus understanding this regulatory cascade may be of importance for human health.

## Materials and methods

### Fly stocks and genetics

RNAi lines are from the Vienna Drosophila Resource Centre or the National Institute of Genetics in Japan. Transgenic fly lines were *UAS-dco* [54] and *P[acman]-EGFP-dsh* [55]. *dsh-ECFP* was cloned into an *pArm-polyA* [56], and *DAnkrd49-EGFP* was cloned in *pActP-FRT-polyA-FRT* [56], followed by P-element mediated insertion into flies. *Bdbt-EGFP* was cloned in *pAttB-ActP-FRT-polyA-FRT* [18] and then inserted into the *attP2* landing site.

Mutant alleles are described in FlyBase. *dco$^{j3B9}$* is a hypomorphic loss of function P-element insertion in the 5'UTR [57, 58]. *l(2)35Be$^1$* and *l(2)35Be$^4$* were isolated in an analysis of the *Adh* genomic region region [59] and subsequently found to carry mutations in *CG4140/DAnkrd49* [35]. Both alleles were obtained from the Bloomington *Drosophila* Stock Centre, but *l(2)35Be$^1$* was not used as it was found to complement *Df(2L)ED3*, *Df(2L)ED800* and *Df(2L)BSC254* that uncover the *l(2)35Be* locus. *l(2)35Be$^4$* however failed to complement all three deficiencies as expected. The *l(2)35Be$^4$* chromosome carries additional mutations (full genotype is *Adh$^{UF}$ l(2) 35Be$^4$ Pol32$^{rd-s}$ pr$^1$ cn$^1$*). Sequencing of *l(2)35Be$^4$* revealed H87Y and A156T mutations within the *DAnkrd49* coding region. Homologues were identified using HomoloGene (https://www.ncbi.nlm.nih.gov/homologene), and protein alignments were carried out using Clustal Omega (http://www.ebi.ac.uk/Tools/msa/clustalo/). SIFT analysis on putative homologues was carried out to determine residues likely to be deleterious (http://sift.jcvi.org).

Null alleles of *CG4140/DAnkrd49* and *CG17282/Bdbt* were made by deletion of the entire open reading frame by homologous recombination using the *pRK2* vector (S2B Fig) [60]. 4 independent mutations of *DAnkrd49* were obtained. All were homozygous lethal and lethal over *DAnkrd49$^{l(2)35Be4}$*, *Df(2L)ED3* and *Df(2L)ED800*. A single mutation in *Bdbt* was obtained: *Bdbt$^{Δ79.1}$* was homozygous lethal, and failed to complement *Df(3R)BSC508* and *Df(3R) ED10845* that uncover the *Bdbt* locus. The phenotypes of *Ankrd49* and *Bdbt* null mutations were rescued by transgenes expressing the appropriate gene.

Mitotic clones were induced using the FLP/FRT system and *Ubx-FLP*. Overexpression of UAS lines or RNAi lines used the GAL4/UAS system with *MS1096-GAL4* [61], *ptc-GAL4* [62] or *en-GAL4* [63].

## Immunostaining and imaging

Adult wings were dehydrated in isopropanol and mounted in GMM (50% methyl salicylate, 50% Canada Balsam).

Unless otherwise indicated, pupal wings were dissected at 28 hr after puparium formation (APF) at 25˚C, as previously described [56]. Primary antibodies for immunostaining were mouse monoclonal anti-Fmi (Flamingo #74, DSHB) [64], rat anti-Dsh [24], affinity purified rabbit anti-Fz [65], rabbit anti-Stbm [15], affinity purified rat anti-Pk [18], affinity purified rabbit anti-GFP (ab6556, Abcam), mouse monoclonal anti-β gal (DSHB), rabbit anti-β gal (#55976, Cappel), rat monoclonal anti-Ecad (DSHB) and mouse monoclonal anti-Arm N2 7A1 (DSHB). Actin was labelled with Alexa568-conjugated Phalloidin (A12380, Molecular Probes).

Pupal wings were imaged on a Nikon A1R GaAsP confocal microscope using a 60x NA1.4 apochromatic lens. 9 Z slices separated by 150 nm were imaged, and then the 3 brightest slices around cell junctions were selected and averaged for each channel in ImageJ. Membrane masks were generated in Packing Analyzer [66]. Polarity magnitude (maximum asymmetry ratio on a cell-by-cell basis) and the variation in polarity angle were calculated as previously described [55]. Values were compared using one-way ANOVA with Dunnett's multiple comparisons test, or using paired t-tests for clones, comparing control and mutant regions of the same wings.

## Biochemistry

For pupal wing westerns, 28 hr APF pupal wings were dissected directly into sample buffer. One pupal wing equivalent was used per lane. Westerns were probed with affinity purified rabbit anti-Dsh [24] and Actin AC-40 mouse monoclonal (A4700, Sigma). Detection was using SuperSignal West Dura Extended Duration Substrate (Thermo Scientific). A BioRad Chemi-Doc XRS+ was used for imaging, and band intensities from six biological replicates were quantified using ImageJ. Data were compared used unpaired t-tests.

For immunoprecipitations, cDNAs were tagged with EGFP or Myc, and cloned into the pAc5.1-V5/His vector (Invitrogen). Immunoprecipitations between Bdbt-EGFP and Myc-DAnkrd49 used GFP rabbit serum (ab290, Abcam), in 50 mM Tris-HCl pH 7.5, 150 mM NaCl, 1% TritonX-100, 1x protease inhibitor cocktail (#11697498001, Roche). Immunoprecipitations between Myc-DAnkrd49 or Myc-Bdt and Dsh-ECFP used Myc antibody resin (ab1253, Abcam), in RIPA buffer (50 mM Tris-HCl pH 7.5, 100 mM NaCl, 1% NP-40, 0.5% sodium deoxycholate, 0.1% SDS, 1x protease inhibitor cocktail). Westerns were probed with mouse monoclonal anti-GFP JL8 (#632381, Clontech) or mouse monoclonal anti-Myc 9E10 (DSHB).

## Supporting information

**S1 Fig. RNAi knockdown of *DAnkrd49* and *Bdbt* in the *ptc-GAL4* expression domain.** (A-C) Dorsal surface of adult male wings from control wild-type flies (A), or from flies expressing RNAi against *CG4140/DAnkrd49* (B, NIG line *4140R-2*), or *CG17282/Bdbt* (C, NIG line *17282R-2*), under control of the *ptc-GAL4* driver at 25˚C, in the presence of *UAS-Dcr2*. Scale bar 100 μm. (D-F) Pupal wings expressing a control RNAi (D, VDRC line *39864*, targeting *Sik1*, a gene unrelated to planar polarity), or RNAi against *CG4140/DAnkrd49* (E, NIG line

*4140R-3*) or *CG17282/Bdbt* (F, NIG line *17282R-2*), under control of the *ptc-GAL4* driver (as in Fig 1E–1G),. Wings immunolabelled for Fmi. The yellow bar shows the *ptc-GAL4* expression domain. The polarity nematic for each cell is shown as red lines. Polarity magnitude (length of red line) is quantitated in the *ptc-GAL4* expression domain, and is reduced in wings expressing RNAi against *DAnkrd49* and *Bdbt* (E,F) compared to control wings (D). The polarity angle is also more regular in wild-type wings.
(TIF)

**S2 Fig. DAnkrd49 conservation and generation of knockout alleles.** (A) Clustal alignment of *DAnkrd49* with homologues. Asterisks (*) indicate conserved residues (also in red), colons (:) indicates conservation between groups of strongly similar properties, and full stop (.) indicates conservation between groups of weakly similar properties. Blue shading indicates the position of the ankyrin repeats, and blue arrows point to residues mutated in *DAnkrd49^{l(2)35Be}*. (B) Diagram showing the open reading frames (blue) in the genomic regions surrounding *DAnkrd49* (top) and *Bdbt* (bottom). Homology arms corresponding to approximately 3 kb on either side of the target gene were inserted into the *pRK2* vector on either side of a *white* gene flanked by LoxP sites (green). Recombination between the *pRK2* transgene and the genomic DNA (red dashed lines) results in the target gene being exchanged for *white* (Δ in red). Regions of the homology arms encoding open reading frames in *pRK2* were sequenced, to ensure no additional mutations were introduced.
(TIF)

**S3 Fig. Protein localisation in *DAnkrd49* and *Bdbt* clones, and *DAnkrd49* knockout mutations.** (A-C) 28hr APF pupal wings carrying clones of cells lacking *DAnkrd49* (A, *l(2)35Be^4*) or *Bdbt* (B and C, *Bdbt^{A79.1}*), marked by loss of β-gal (green). Wings immunolabelled for Stbm (red) and Arm (blue). (A,B) Arm staining is similar to wild-type in *DAnkrd49* clones and some *Bdbt clones*. (C) In other *Bdbt* clones cells are abnormal, as seen by disrupted Arm localisation. Scale bar 10 μm. (D-F) 28hr APF pupal wings carrying clones of cells lacking *DAnkrd49* (*DAnkrd49^{ΔD3.1}*), marked by loss of β-gal (red in D and E or blue in F). Wings immunolabelled for Fmi (green in D and F), Dsh (green in E) and Phalloidin (red in F).
(TIF)

**S4 Fig. Fz, Pk and Dsh localisation in *DAnkrd49* and *Bdbt* mutant clones.** (A-D) 28hr APF pupal wings carrying clones of cells lacking *DAnkrd49* (A,C, *DAnkrd49^{l(2)35Be4}*), or *Bdbt* (B,D, *Bdbt^{A79.1}*) marked by loss of β-gal (red). Wings immunolabelled for Fz (green in A,B), or Pk (green in C,D). Scale bar 10 μm. (E,F) 28hr APF pupal wings carrying clones of cells lacking *DAnkrd49* (E, *DAnkrd49^{l(2)35Be4}*), or *Bdbt* (F, *Bdbt^{A79.1}*), in a background expressing one copy of *Arm-dsh-ECFP*. Wings immunolabelled for GFP (green), clones marked by loss of β-gal (red). (G,H) 28hr APF pupal wings carrying clones of cells lacking *DAnkrd49* (*DAnkrd49^{l(2)35Be4}*) in a background expressing *ActP-DAnkrd49-EGFP* (G), or clones of cells lacking *Bdbt* (*Bdbt^{A79.1}*) in a background expressing *ActP-Bdbt-EGFP* (H). Wings immunolabelled for GFP (green) and Dsh (red). Clones marked by loss of β-gal (blue). Dsh levels within the clone are similar to wild type in the presence of DAnkrd49-EGFP or Bdbt-EGFP (compare Dsh labelling inside and outside the clone tissue).
(TIF)

**S5 Fig. Immunoprecipitation experiments with DAnkrd49/Bdbt and Dsh.** (A) Western blot showing co-immunoprecipitation experiment. S2 cells transfected with Dsh-ECFP and either Myc-DAnkrd49 or Myc-Bdbt. Immunoprecipitation with Myc antibody resin did not pull down Dsh-ECFP.
(TIF)

**S1 Table. DAnkrd49 and Bdbt adult wing phenotypes.** Lines targeting CG4140 are 4140R-2 and 4140R-3, from the NIG RNAi collection, and 26396 and 109913 which are GD and KK lines respectively, from the VDRC. 4140R-2 and 4140R-3 are two insertions of the same target sequence, and target sequences overlap with lines 26396 and 109913. Lines targeting CG17282 are 17282R-2 and 17282R-3 from the NIG RNAi collection, and GD line 40059 from the VDRC. 17282R-2 and 17282R-3 are two insertions of the same target sequence, and target sequences overlap with line 40059.
(DOCX)

**S2 Table. Numerical data for Figs 3, 4, 5 and 7.**
(XLSX)

## Acknowledgments

We thank Mike Young, the Bloomington Drosophila Stock Center, Vienna Drosophila Resource Center and the National Institute of Genetics Japan for fly stocks, and the Developmental Studies Hybridoma Bank for antibodies. cDNAs were provided by the Drosophila Genomics Resource Centre and P[acman] constructs by BacPac Resources. BestGene and Genetivision are thanked for making transgenics, Rosalind Hale for making the *Bdbt* mutant and Victoria Thomas-MacArthur for excellent support for the RNAi screen. Amy Brittle, Simon Fellgett and Katie Fisher are thanked for comments on the manuscript, and the fly room staff for technical support. Imaging was performed in the Wolfson Light Microscopy Facility.

## Author Contributions

**Conceptualization:** Helen Strutt, David Strutt.

**Funding acquisition:** David Strutt.

**Investigation:** Helen Strutt.

**Writing – original draft:** Helen Strutt.

**Writing – review & editing:** Helen Strutt, David Strutt.

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
