## [Decision Letter · Decision Letter 0]

24 May 2020

Dear Dr Strutt,

Thank you very much for submitting your Research Article entitled 'DAnkrd49 and Bdbt act via Casein kinase Iε to regulate planar polarity in Drosophila' to PLOS Genetics. Your manuscript was fully evaluated at the editorial level and by independent peer reviewers. The reviewers appreciated the attention to an important topic but identified some aspects of the manuscript that should be improved.

We therefore ask you to modify the manuscript according to the review recommendations before we can consider your manuscript for acceptance. Your revisions should address the specific points made by each reviewer.

[LINK]

Yours sincerely,

Tadashi Uemura

Guest Editor

PLOS Genetics

Gregory P. Copenhaver

Editor-in-Chief

PLOS Genetics

Dear Helen and David,

As you see, the reviewers are positive and gave several comments, which I consider are reasonable. Please attend to all of the reviewer’s comments carefully. I would like to encourage you to perform additional experiments as the reviewers suggested, except for comment 5 of reviewer 2. Even if results of those experiments are negative, please state those and discuss why they happen. If you are able to revise the manuscript in principle along the lines suggested, I will be happy to consider it swiftly.

I hope that both of you are healthy and in good spirits even in this unprecedented and still unforeseeable situation.

With best regards,

Tadashi Uemura

Reviewer's Responses to Questions

**Comments to the Authors:**

Reviewer #1: In this manuscript, Strutt et al. identified Bdbt and DAnkrd49 as novel planar cell polarity (PCP) proteins. The authors revealed that these two proteins physically interact in vitro and regulate each other’s protein levels in vivo. They also found that Bdbt and DAnkrd49 are essential for asymmetric localization of the core proteins in the Drosophila pupal wing and regulate overall levels and phosphorylation of Dsh. Further analyses provided evidence that Bdbt and DAnkrd49 act together to regulate Dco activity, which in turn phosphorylates Dsh during PCP establishment.

Overall, the paper is clearly written and most of the experiments are well designed. The findings in this study provide an important contribution to understanding the molecular mechanism of PCP. However, I think that the following points, especially the molecular mechanism on how DAnkrd49 regulates Dco activity, need to be clarified and/or corrected.

(1) A previous study by Fan et al. demonstrated that in the context of circadian rhythms, Bdbt physically interacts with Dco and positively regulates its activity. The authors and other groups previously reported that Dco phosphorylates Dsh in PCP signaling. These data taken together propose a model that Bdbt modulates Dco activity, which in turn regulates Dsh phosphorylation. In this study, the authors demonstrated that this regulatory pathway actually works during PCP establishment in the Drosophila wing and that DAnkrd49 is also implicated in this pathway through the regulation of Dco activity. However, it is unclear to me how DAnkrd49 contributes to the regulation of Dco activity. Obviously, one possible explanation is that DAnkrd49 regulates Dco activity through interaction with and stabilization of Bdbt. The authors should examine whether the two point mutations seen in the DAnkrd49l(2)35Be4 allele, which causes a reduction of Bdbt in vivo, affect the physical interaction between DAnkrd49 with Bdbt.

(2) This is related to point 1 above. It would be important to examine whether DAnkrd49, Bdbt and Dco can form a trimeric complex.

(3) In Figure1, control pictures should be provided for comparison.

(4) In Figure 3A’, the control region on the bottom-left corner seems to be treated as a part of the mutant clone (because of no yellow line). If so, the values should be corrected.

(5) Page 7, line 147: “target genes” should be “target proteins”.

Reviewer #2: In this study, the authors identified two new regulators of drosophila PCP. They found Bdbt and dAnkrd49 bind to each other to regulate Dsh protein levels and Dsh phosphorylation via Ser/Thr kinase Dco. The manuscript is well written and the data is of good quality and convincing. It provides new insights into the regulatory mechanism of core PCP proteins. It is suitable for publication in PLoS Genetics after the following comments and suggestions have been addressed.

1. Does Ankrd49 directly bind to Dco, or it affects Dco indirectly through stabilizing Bdbt?

2. Why is the phenotype mainly restricted to the proximal region of the wing? What is the normal expression pattern of Bdbt and dAnkrd49 in drosophila wing?

3. Figure 4A, B, the rescue effect of DAnkrd49-EGFP and Bdbt-EGFP should also be shown by the trichome orientation.

4. Authors found it is difficult to verify whether loss of Dankrd49 or Bdbt reduces Stbm phosphorylation due to the limited mobility shift of Stbm on the SDS-PAGE. Is it possible to overexpress Fz to induce Stbm phosphorylation (as shown by Kelly et al. Cell Reports 2016) to make it more visible, then test the effect of Dankrd49 or Bdbt?

5. In mammalian cells, Dvl is also highly phosphorylated by CK1. Given the potential interaction between Ankdr49, CK1 and FKBP family members, does knockdown or knockout of Ankdr49 (or FKBP family members) in mammalian cells also affect Dvl phosphorylation and protein levels? Although it is not necessary for this paper, I would suggest this experiment because it will increase the interest and impact of the manuscript for readers working in mammalian system.

6. A discussion regarding the phenotype of increase of cell size (Figure 1) and lethality would be useful. Is it because of additional substrates of Dco?

**Have all data underlying the figures and results presented in the manuscript been provided?**

Reviewer #1: Yes

Reviewer #2: Yes

PLOS authors have the option to publish the peer review history of their article (what does this mean?). If published, this will include your full peer review and any attached files.

Reviewer #1: No

Reviewer #2: No

---

## [Decision Letter · Decision Letter 1]

19 Jun 2020

Dear Helen,

We are pleased to inform you that your manuscript entitled "DAnkrd49 and Bdbt act via Casein kinase Iε to regulate planar polarity in Drosophila" has been editorially accepted for publication in PLOS Genetics. Congratulations!

Yours sincerely,

Tadashi Uemura

Guest Editor

PLOS Genetics

Gregory P. Copenhaver

Editor-in-Chief

PLOS Genetics

Comments from the reviewers (if applicable):

Congratulations!

Reviewer's Responses to Questions

**Comments to the Authors:**

Reviewer #1: The revised manuscript by Strutt and Strutt has been adequately improved. The authors have addressed the issues that I raised in my previous review by adding the control pictures for comparison and correcting the errors. They also further discussed the relationship among DAnkrd49, Bdbt and Dco using the additional data, which strengthened their conclusions. I am satisfied with the revised version.

Reviewer #2: My concerns have been addressed satisfactorily, a nice paper.

**Have all data underlying the figures and results presented in the manuscript been provided?**

Reviewer #1: Yes

Reviewer #2: Yes

PLOS authors have the option to publish the peer review history of their article (what does this mean?). If published, this will include your full peer review and any attached files.

Reviewer #1: No

Reviewer #2: No

**Data Deposition**

http://datadryad.org/submit?journalID=pgenetics&manu=PGENETICS-D-20-00651R1

**Press Queries**

---

## [Editor Report · Acceptance letter]

16 Jul 2020

PGENETICS-D-20-00651R1 

DAnkrd49 and Bdbt act via Casein kinase Iε to regulate planar polarity in <I>Drosophila</I> 

Dear Dr Strutt, 

We are pleased to inform you that your manuscript entitled "DAnkrd49 and Bdbt act via Casein kinase Iε to regulate planar polarity in <I>Drosophila</I>" has been formally accepted for publication in PLOS Genetics! Your manuscript is now with our production department and you will be notified of the publication date in due course.

With kind regards,

Kaitlin Butler

PLOS Genetics

On behalf of:
